# Evaluating source credibility effects in health labelling using vending machines in a hospital setting

**Melda Griffiths[1], Jacky Boivin[1], Eryl Powell[2], Lewis Bott** [1] *

**1** School of Psychology, Cardiff University, Cardiff, Wales, United Kingdom, **2** Aneurin Bevan Gwent Public Health Team, Newport, Wales, United Kingdom

\* bottla@cardiff.ac.uk

## Abstract

### Objectives

Providing advice to consumers in the form of labelling may mitigate the increased availability and low cost of foods that contribute to the obesity problem. Our objective was to test whether making the source of the health advice on the label more credible makes labelling more effective.

### Methods and measures

Vending machines in different locations were stocked with healthy and unhealthy products in a hospital. Healthy products were randomly assigned to one of three conditions (i) a control condition in which no labelling was present (ii) a low source credibility label, "Lighter choices", and (iii) a high source credibility label that included the UK National Health Service (NHS) logo and name, "NHS lighter choices". Unhealthy products received no labelling. The outcome measure was sales volume.

### Results

There were no main effects of labelling. However, there were significant interactions between labelling, vending machine location and payment type. For one location and payment type, sales of products increased in the high credibility label condition compared to control, particularly for unhealthy products, contrary to expectations.

### Conclusions

Our findings suggest that high source credibility health labels (NHS endorsement) on food either have little effect, or worse, can "backfire" and lead to effects opposite to those intended. The primary limitations are the limited range of source credibility labels and the scale of the study.

**Data Availability Statement:** Dataset is publicly available on the Open Science Framework, DOI: DOI 10.17605/OSF.IO/HYWDB.

**Funding:** This work was supported by the Economic and Social Research Council, grant

number ES/J500197/1, and the Aneurin Bevan University Health Board, grant number 512976. The funders had no role in study design, data collection and analysis, decision to publish, or preparation of the manuscript.

**Competing interests:** The authors have declared that no competing interests exist.

# Background

Obesity remains as one of the 21st century's most significant challenges to public health within the developed and developing world [1, 2]. Current health systems are failing in their efforts to achieve and maintain healthy weights and healthy energy balances within our populations, with no evidence of successful efforts to turn the tide and decrease overweight and obesity prevalence levels in any nation [3]. Chronic positive energy balance, where the energy consumed consistently outweighs the energy expended, results in overweight and obesity, which in turn are linked to numerous non-communicable diseases, physical disabilities and psychological issues [2, 3]. Tobacco, alcohol and physical inactivity combined produce a smaller non-communicable disease burden than that presented by poor diets [4]. Therefore, in addressing the obesity epidemic, measures that are supportive of healthier diets are essential.

More responsible food marketing could play a role in making diets healthier. As the role of food marketing in making food environments obesogenic has become better understood, the appetite for marketing the healthiness of food to promote healthy sales has also grown [5]. One approach that aligns with this goal is the use of health labels. Providing information at the point-of-purchase that eases the identification of healthier options could ease the selection and thus the consumption of healthier options [6]. However, while some studies have found robust, positive effects of labelling, particularly when combined with other interventions [7–9], many other studies have found only small or non-existent effects [10–12]. Although the explanation for the variability in effectiveness of labelling is likely to be multifaceted, one important consideration is whether the messages motivate consumers sufficiently for them to change their behavior [13]. In this study, we test whether highlighting a trusted source of health advice alongside health information on a label provides the additional motivation necessary for positive behavior change.

## Source effects

Research in social psychology has demonstrated that the source of a message can have an impact on how it is processed. For example, one study demonstrated that source likeability influences the extent to which participants' opinions change to align with that of the source [14]. It has been argued that individuals are more susceptible to persuasion if it comes from a credible source (with credible sources holding expertise, being trustworthy and being good willed) [15]. In the context of sales, if claims about products come from trusted sources, the claims are more likely to be accepted [16]. People often defer to a course of action recommended by someone with expertise [17] and knowing that a message comes from a trusted source works as a mental shortcut to simplify the decision-making process. As such, using trusted sources to endorse claims makes these claims more credible to the consumer [16, 18–21]. Supporting health claims on health labels with an endorsement from a credible source could therefore be an effective mental shortcut for consumers to rely upon when selecting healthier foods.

There is little direct evidence demonstrating the impact of source credibility on the effectiveness of food health labels in shaping consumer behavior. However, available indirect evidence is consistent with higher credibility sources having positive effects [19, 22–24]. In their study, Feunekes and colleagues tested whether endorsements by international and national health organizations have an influence on evaluations of health label credibility [23]. In an online study with over 1,600 participants across four European countries, smiley face health labels were presented with an endorsement from either the World Health Organization (WHO), the European Union (EU), European Food Manufacturers (EFM), or a national nutrition organization from the relevant country. Results showed that credibility was significantly

higher in the presence of the endorsement. Furthermore, the less nutritionally targeted EU and EFM endorsements were evaluated as less credible than those from the WHO and the respective national organizations.

While these results suggest there are benefits to high source credibility labelling, they are limited by the use of attitude change as a measure rather than behavior. Source credibility might alter attitudes and intentions but not behavior. Without further insights on the impact health labels with credible sources have on purchasing behavior, it is difficult to estimate how effective they might be as an intervention, and as such, how cost effective such interventions might be.

In summary, health labels may serve as an easy and affordable method of nudging consumer behavior but they have shown only limited effectiveness. Attributing the labelling to a trusted, highly credible source of health information might boost the effect of the label. In this study, we test source credibility directly using purchasing behavior as the outcome measure.

## Current study

We tested whether high source credibility improved labelling using vending machines. Vending machines are a useful tool for studying the effects of interventions on food choice as they offer exceptional levels of experimental control [8, 25, 26]. Sales data were collected from two vending machines at a hospital site over three months, with one machine in the A&E department and another at reception. The machines sold healthy products (e.g., baked crisps, cereal bars) and unhealthy products (e.g., standard crisps, chocolate bars) in equal proportions. Each week, the machines were independently and randomly assigned to one of three conditions, with the outcome measure being the number of products sold. There were two labelling conditions and a control, no label condition. In the low credibility condition, healthy products were labelled "lighter choices" (Fig 1). In the high credibility condition, the same label was appended with the National Health Service (NHS) logo ("NHS lighter choices"; Fig 2). In the UK, the NHS is one of the most trusted sources of health information, and its endorsement is often seen as a stamp of approval for products or treatments [27, 28].

If high source credibility improves the effects of labelling, purchasing behavior should shift from unhealthy products to healthy products. There should therefore be more healthy purchases and fewer unhealthy purchases in the high source credibility condition.

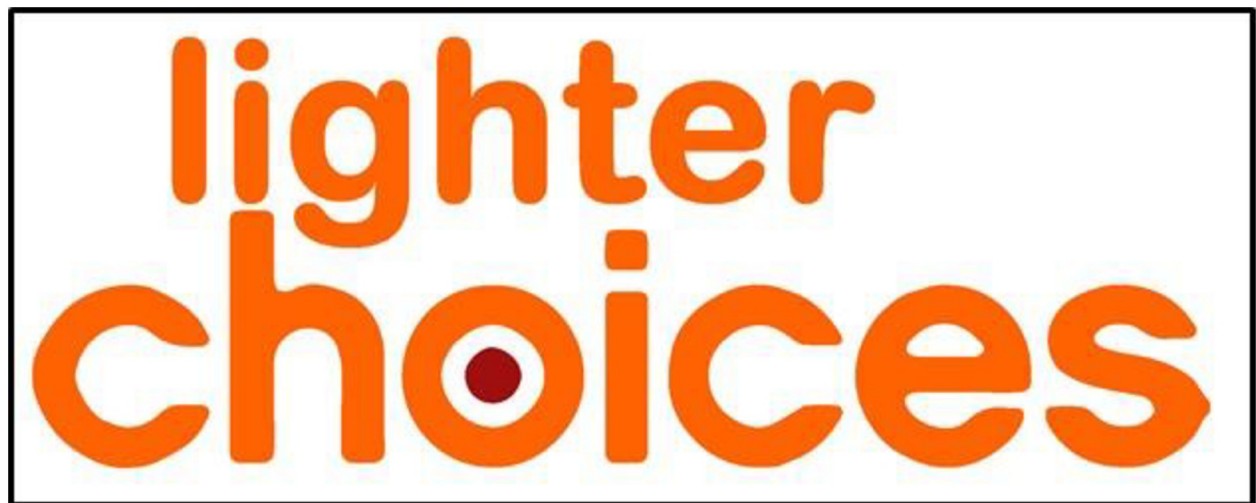

**Fig 1. Low credibility label.**

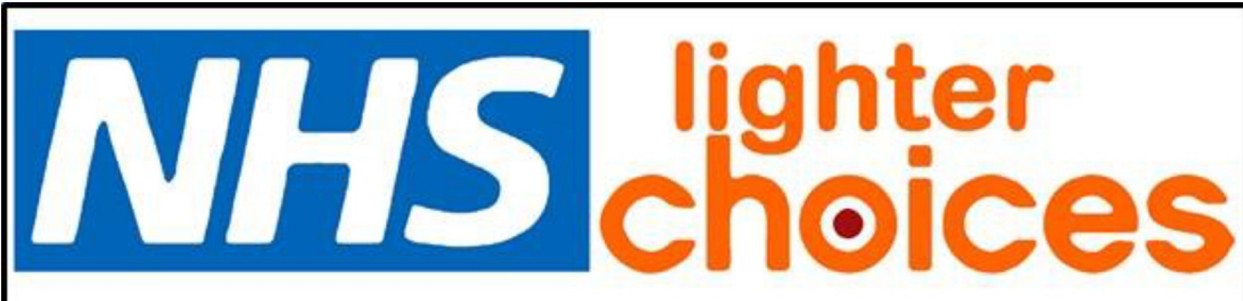

**Fig 2. High credibility label.**

## Methods

### Participants

Participants voluntarily made purchases from vending machines without being aware that they were taking part in a study. Ethical approval was granted from the NHS research ethics committee (IRAS number: 231390) and the Cardiff University School of Psychology Ethics Committee. Permission was given not to collect consent from participants and to not inform them of the nature of the study.

### Study design

Two vending machines were used (see Implementation in S1 Methods). One machine was in Accident and Emergency (A&E), the other in a reception area. Location and number of vending machines was determined by practical rather than scientific considerations.

The study lasted 90 days. Every six days, each machine was randomly allocated to one of the labelling conditions (no label, low credibility, high credibility) such that each machine was in each condition five times in total. Practical reasons limited the length of the study to 90 days rather than power considerations.

Machines housed the same stock in the same positions within the machines throughout. There were 12 healthy products and 11 unhealthy products (see Vending in S1 Methods). Healthy snacks satisfied the Welsh Hospital Healthy Vending directive constraints [29] whereas unhealthy products did not. All products were sold for the same price (£1).

Purchasers could use either cash or credit card in the vending machines.

### Labels

Labels were placed underneath the product referred to (see Labels in S1 Methods).

### Analysis

The data were analysed as a mixed model with the lme4 package in R [30]. Sales volume was the outcome variable e.g., to evaluate whether the high credibility label influenced purchases we compared total sales in the high credibility condition against total sales in the no label condition. Individual products (N = 23) and time (N = 15) were included as random effects. This allowed for generalisation across time and product range. Model specification was maximal [31], in that all possible random effects parameters were included.

We computed an omnibus regression model that included labelling (no label, low credibility, high credibility), product health value (healthy, unhealthy), machine (A&E, reception),

and payment type (cash, credit card) as fixed effects, and product and time as random effects. The model is shown below in R pseudo-code:

**Omnibus model.**

```
salesvolume ~ label * healthvalue * machine * payment +
(1+condition*machine|item) + (1+machine*healthvalue|time))
```

The reference levels were "unhealthy" for health value, "A&E" for machine, and "no label" for credibility label. Sum coding was used throughout. We also analysed subsets of the data by reducing the omnibus model appropriately e.g., remove the machine factor.

An alternative analysis strategy would be to assume a binary outcome measure in which each participant purchased either a healthy or an unhealthy product. The data would then be analysed as a logistic regression with predictors of label, machine and payment type as fixed effects and participant, product type, and time as random effects. However, such a model fails to take into account that the different labels might attract different numbers of purchases e.g., participants might make more purchases in the high credibility condition than the low credibility condition. A logistic regression model assumes that the number of data points is independent of the fixed effect level but in our study, there is a realistic possibility that the number of data points (purchases) is influenced by the fixed effects. Using sales volume as the outcome measure takes this into account.

## Results

There were 10734 sales in total, of which 6465 were from the A&E machine and 4269 were from the Reception machine. There were 2633 healthy sales and 8101 unhealthy sales. Cash accounted for 7235 sales and credit card for 3499 sales.

Sales by labelling condition are shown in Fig 3. Significantly more unhealthy sales were made than healthy sales, $\beta = 4.31$, se = 0.86, t = 5.00, p < .001, and there were more sales in the A&E machine than in the reception machines, $\beta = 1.62$, se = 0.29, t = 5.53, p < .001. There were also more sales by cash than credit card, $\beta = 2.78$, se = 0.097, t = 28.53, p < .001.

Interactions with machine, health value and payment type were also present. Although both machines saw greater sales of unhealthy products than healthy products, the difference was greater in the A&E machine than the reception machine, $\beta = 0.69$, se = 0.28, t = 2.46, p =

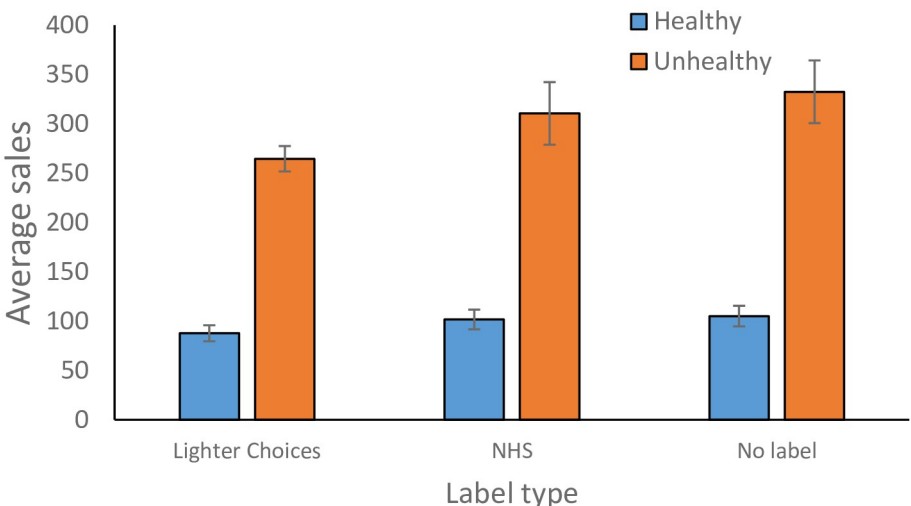

**Fig 3. Average number of product sales per six-day time-period.** There were significantly more unhealthy sales than healthy sales but no main effect of labelling. Error bars correspond to the standard error of the mean across items.

.022. Similarly, both machines saw greater cash sales than credit card sales but again the difference was greater in the A&E machine, β = 0.50, se = 0.097, t = 5.09, p < .0001. Finally, there was an interaction of health value by payment type. Cash and credit card purchases experienced more unhealthy than healthy sales but the difference was greater for cash than credit card sales, β = 1.64, se = 0.097, t = 16.88, p < .000. However, it is important to note that proportions of unhealthy and healthy sales stayed approximately constant across machines and payment type: healthy sales accounted for 26% of purchases in the A&E machine and 23% in the reception machine; and 24% of purchases in cash sales and 26% in credit card sales. The significant interactions arise because there were more sales overall in the A&E machine and in cash, and the model tests for absolute differences not proportion differences.

There were no main effects of labelling on sales. The low credibility label condition did not differ significantly to the no label condition, β = -0.15, se = 0.21, t = -0.72, p = 0.48, nor did the high credibility label differ to the no label condition, β = -0.14, se = 0.20, t = -0.68, p = 0.50., and the low credibility label did not differ to the high credibility label, β = 0.28, se = 0.20, t = 1.43, p = 0.17.

However, there were significant three-way interactions including label. There was a significant label by machine by payment type interaction involving no label and high credibility label, β = 0.41, se = 0.14, t = -2.96, p = 0.0031 and a significant label by machine by payment type interaction involving low credibility and high credibility label, β = 0.32, se = 0.14, t = 2.35, p = 0.019, but no significant label by machine by payment type interaction involving low credibility and high credibility label, β = 0.084, se = 0.14, t = 0.61, p = 0.54. Fig 4 illustrates that these effects arose because of high sales in the high credibility label, credit card payment, A&E machine. There was also a marginal four-way interaction between label (no label *vs* high credibility), health value, machine, and payment, β = 0.25, se = 0.14, t = -1.79, p = 0.073.

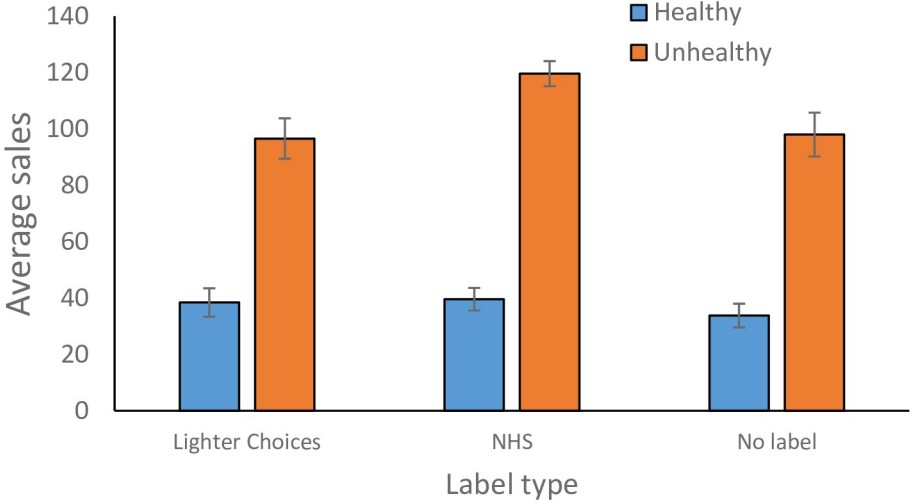

**Fig 4. Average number of product sales per six-day time-period for A&E machine, credit card sales only.** For the A&E machine, credit card sales, using the high source label (NHS) yielded significantly more unhealthy sales than either the lighter choices or no label conditions, contrary to expectations. Error bars correspond to the standard error of the mean across items.

To investigate these effects further, we analysed each machine separately, for each payment type. In all four analyses, there were more unhealthy products sold than healthy products, all t's > 4.29, all p's < .001. However, only the A&E machine, credit card sales, showed effects of labelling. Here, the high credibility label resulted in significantly more purchases than the no label condition, β = 0.77, se = 0.35, t = 2.24, p = 0.043, but no more than the low credibility condition, β = -0.32, se = 0.36, t = -0.89, p = 0.39. The low credibility purchases did not differ to the no label condition, β = -0.45, se = 0.34, t = -1.309, p = 0.21.

There was also evidence that the high credibility label increased sales of unhealthy products more than healthy products in the A&E machine, credit card sales. The interaction between health value and label (no label *vs* high credibility) was significant, β = 0.58, se = 0.26, t = 2.20, p = 0.040, consistent with the marginally significant four-way interaction from the omnibus analysis. However, neither of the other health value by label interactions (no label *vs* low credibility, low credibility *vs* no label) were significant, t's < 1.48, p's > 0.16.

Overall, we found no evidence that high credibility source labelling improved the effectiveness of healthy labelling. Indeed, there was evidence that high credibility labels increased sales of unhealthy products.

## General discussion

A key challenge for using health labels is ensuring that their messages motivate consumers sufficiently for them to change their behavior for the better [13]. Health advice alone may not be sufficient to motivate individuals to use health labels and opt for healthier options. Harnessing heuristic processing could improve the chances of a label being used, and for it to be influential. Previous research has shown that messages from trusted sources, such as health organizations, are more likely to be viewed as credible. This in turn boosts their persuasiveness, increases the extent to which their claims are accepted by consumers, and ultimately improves product evaluations [11, 15, 16, 18–21, 23, 32]. In this study, we aimed to establish whether health labels could be made more effective in promoting healthier food purchases by the inclusion of a high credibility source of health information (NHS endorsement) on the label. Our results do not provide evidence that they do. Indeed, high credibility labels promoted more sales in the vending machine located in A&E when purchased by credit card, and more unhealthy sales in particular. Potential explanations and its implications for public health are discussed below.

### Backfire effect

We found that the high source credibility label increased sales. The effect was restricted to the A&E machine and to credit card sales, but the effect was nonetheless significant. An intervention designed to reduce consumption that leads to an increase in consumption, albeit under restricted conditions, is the reverse of the intended result, i.e., *a backfire effect*. We consider three explanations for this.

The first is that there was psychological reactance against the label [33]. Theories of psychological reactance postulate that imposing rules or constraining freedoms can result in individuals reacting against them and showing increased desire for the restricted choice [33]. While health labels do not enforce choices, they suggest courses of action, which some may interpret as an intrusion of their personal freedom [34]. There is evidence of reluctance to accept health claims by health promoters [35], and attempts to influence diets are sometimes perceived as paternalistic violations of agency and freedom of choice by the government [36]. In the context of this study, consumers might have objected to the suggestion that they should be making certain food choices [33, 34], with the high credibility label being perceived as particularly

paternalistic. An alternative possible source of reactance is the fact that individuals in A&E may have been waiting a long time to receive medical attention. They may have felt irritated with the NHS about their predicament and reacted by deliberately choosing the products not endorsed by the NHS.

The second explanation for the backfire effect is based on goal fulfillment. Evidence suggests that the presence of a healthier option within a selection of less healthy options can increase unhealthy consumer behavior [37]. In their study, Wilcox and colleagues showed that high self-control individuals were more likely to select the most indulgent side option available within a range when the range included a salad option, compared to when it did not. The authors hypothesized that exposure to healthier items vicariously satisfied consumers' existing health goals (regardless of whether the product was chosen) [37]. The presence of NHS labelling may have indirectly supported goal fulfillment. Making a selection from a range that had a strong NHS branding presence could have been interpreted as making sufficient progress towards consumers' health goals.

The final explanation is that the labelling may have negatively influenced product perceptions. While for some, healthier foods are more attractive, there is evidence that healthy foods can be perceived as less tasty, less satiating, and more expensive [38–40]. The NHS label may have drawn attention to the healthy features of the products, and in doing so, inadvertently made the products appear less attractive for consumption, e.g., less tasty. Future studies can investigate which sorts of healthy messaging give rise to negative product perceptions and whether the labels used here fall into that category.

## Variability across location and payment type

Backfire effects were larger in the A&E machine than the reception machine. Why was there such a difference between the two locations? We consider two possibilities. First, if consumers were reacting against the label, they would have been more likely to do so in A&E than in reception. Consumers would have been more stressed in A&E, they may have been waiting for a long time, concerned about those admitted to hospital, hungry or tired. These factors are likely to maximize irritation with the authority who they viewed responsible for their wait (the NHS).

Second, the effects of labelling more generally may be greater in A&E than in the reception area. Consumers in A&E are captive–many are waiting with little to occupy themselves, allowing time for using the labels and absorbing their content. In comparison, the reception area had high passing footfall and minimal seating. Participants in the reception area may not have attended to the labels to the same degree as those in A&E. The difference in mental states of consumers between the two locations might also contributed to greater use of the label in A&E. Mental shortcuts are more likely in scenarios where individuals wish to exert minimal cognitive effort when making their choices [14, 41, 42] and indeed stress has been shown to increase reliance on mental shortcuts [43]. Participants in A&E may have sought to ease the decision process as much as possible by using cues available within the environment to help them make a selection.

There was also variability across payment type. Participants paying by credit card were influenced by the labels to a greater extent than those paying by cash (there was no evidence that cash purchases were influenced by the label). It is not clear why. Perhaps those who made multi-purchases (presumably over-represented in credit card payments) were more likely to use the label as a source of information because they had to make more choices within the fixed range. Multi-purchases might also have been a sign of greater stress and greater irritation with the environment, and therefore reactance was more likely.

## Limitations and future work

One limitation of this study was that it used only a single type of high credibility label—the NHS endorsement. While the NHS is a trusted source of health messaging [27, 28], it is possible that endorsement by the NHS may not have provided the credibility that we assumed. At least two issues could be problematic. First, the NHS not only provides health advice, but it also manages health sites such as hospitals. It can therefore be seen as responsible for the predicaments of the consumer, e.g., long waiting times in A&E. A high credibility label divorced from the management of the health site may not lead to the same negative effects as the NHS label and may even promote healthier sales. Future research needs to investigate why we observed the negative effects of the NHS label and whether these effects generalise to other high credibility sources.

Second, trust in the NHS may not span across all health-related messaging. It may be most suited towards messaging around health services, rather than messaging that promotes certain dietary behaviours in lieu of others. NHS England commissioned a qualitative investigation of the NHS brand identity, which highlighted how NHS branding was recognisable and brought confidence about the services being delivered [44]. Whether this same confidence would expand beyond NHS services and into social marketing practices that aim to promote healthier behaviors is unknown. Further work should investigate the degree which the NHS is trusted to deliver dietary advice as well as services.

A related issue concerns perceptions of the healthiness of the products. The products sold in the vending machine were selected based on whether they were compliant or non-compliant with Welsh Hospital Healthy Vending directive constraints [29]. However, this does not guarantee that consumers perceived these products as intended (i.e., healthy vs unhealthy). Any incongruency between consumers perceptions of product health, and what is stated on a label, could impact on attitudes and in turn behaviour.

Finally, there were two limitations with respect to experimental design. First, practical constraints limited our work to a single hospital across a thirteen-week time window. A greater sample of sites and longer study would have increased power to detect differences between labels (e.g., French and colleagues tested at 24 sites [8]). Furthermore, the hospital environment might have been unusual in that health is prominent and stress levels are high. Generalising our findings to sites other than hospitals will therefore require widening the range of sites tested.

Secondly, participants were not individually randomly allocated to a labelling condition. Instead, they were randomly allocated as a group, based on the six-day period in which they were at the hospital e.g., those who arrived at A&E in period 7 were allocated to the high credibility condition. This meant that observations (participants) were less likely to be independent than if we had allocated each participant individually to a labelling condition. This issue was partly alleviated by allocating each labelling condition to multiple, randomly selected weeks within the study period, rather than just one, but an improvement would be to randomly allocate at an even finer level of temporal granularity e.g., each day rather than each week.

## Conclusion and implications

The purpose of this study was to determine whether greater source credibility would improve the performance of health advice labelling in promoting healthy snack choices. We found no evidence that it does: There were no positive effects of the labels, with or without high source credibility. Indeed, we observed a significant backfire effect whereby consumers selected a greater proportion of unhealthy products when healthy products were accompanied by a high credibility label, the NHS endorsement, at least for participants who paid by credit card. While

testing with a wider range of high credibility endorsements is needed to establish to what extent our results generalise, the health implications for labelling in hospitals are clear: high source credibility labelling is not effective in high stress environments and may even cause harm.

## Supporting information

**S1 Methods. Supplementary information on methods.**
(DOCX)

## Acknowledgments

The authors wish to thank JDJ Vending Services and the facilities team at the Aneurin Bevan University Health Board (namely Arabella Roberts and Gareth Hughes) for their support and cooperation in conducting this study.

## Author Contributions

**Conceptualization:** Melda Griffiths, Jacky Boivin, Eryl Powell, Lewis Bott.

**Data curation:** Melda Griffiths, Lewis Bott.

**Formal analysis:** Melda Griffiths, Lewis Bott.

**Funding acquisition:** Eryl Powell, Lewis Bott.

**Investigation:** Melda Griffiths, Jacky Boivin, Eryl Powell, Lewis Bott.

**Methodology:** Melda Griffiths, Jacky Boivin, Eryl Powell, Lewis Bott.

**Project administration:** Eryl Powell, Lewis Bott.

**Resources:** Lewis Bott.

**Supervision:** Eryl Powell, Lewis Bott.

**Writing – original draft:** Melda Griffiths, Lewis Bott.

**Writing – review & editing:** Melda Griffiths, Jacky Boivin, Eryl Powell, Lewis Bott.

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
