## [Decision Letter · Decision Letter 0]

28 Jun 2023

PONE-D-23-00303Evaluating source credibility effects in health labelling using vending machines in a hospital settingPLOS ONE

Dear Dr. Bott,

Thank you for submitting your manuscript to PLOS ONE. After careful consideration, we feel that it has merit but does not fully meet PLOS ONE’s publication criteria as it currently stands. Therefore, we invite you to submit a revised version of the manuscript that addresses the points raised during the review process.

We look forward to receiving your revised manuscript.

Kind regards,

Ali B. Mahmoud, Ph.D.

Academic Editor

PLOS ONE

Journal Requirements:

This work was supported by the Economic and Social Research Council, grant number ES/J500197/1, and the Aneurin Bevan University Health Board, grant number 512976. Awards were made to LB.

No authors have competing interests.

Reviewers' comments:

Reviewer's Responses to Questions

**Comments to the Author**

1. Is the manuscript technically sound, and do the data support the conclusions?

Reviewer #1: Partly

Reviewer #2: Yes

2. Has the statistical analysis been performed appropriately and rigorously? 

Reviewer #1: I Don't Know

Reviewer #2: Yes

3. Have the authors made all data underlying the findings in their manuscript fully available?

Reviewer #1: Yes

Reviewer #2: Yes

4. Is the manuscript presented in an intelligible fashion and written in standard English?

Reviewer #1: Yes

Reviewer #2: Yes

5. Review Comments to the Author

Reviewer #1: Introduction

Introduction is clearly written and succinct. A couple of points to consider:

1. The authors contextualise the main content of the introduction by sharing they found evidence for a label effect opposite to the one intended (so-called ‘backfire’ effect, where label credibility appears to increase unhealthy food sales in a particular context, rather than health food sales as intended). The possibility of backfire effects are outlined in the introduction, but it is not clear whether these effects were truly anticipated (or not) when the study was being set up. If they were anticipated, I would make sense to describe this in the introduction, but I would have expected more effort put into characterising how possible consumers are likely to perceive the various labels, prior to them being placed in the vending machines. See also my comment on the methods below. However, if the ‘backfire’ effects where not anticipated when the study was designed, it would be better to focus on this as a critique in the discussion.

2. Related to the above, although there may not be a lot of literature investigating source credibility effects in the food choice domain, there are quite a number of studies attempting to characterise the impact of ‘healthier’/ ‘lighter’/ ‘lower calorie’ labels and ‘healthier’ perceptions on consumer beliefs, including willingness to pay, and food choice, portion decisions and even intake. The body of evidence broadly indicates that ‘healthier’ labels/perceptions can impact consumer behaviours in a number of ways, and not always as intended. For example, ‘healthier’ perceptions are sometimes associated with poorer taste quality, which can reduce consumers interest in selecting these products. It would be good to see some of this consumer-focused literature considered in the paper. The NHS is sited as a trusted source of health information, but is it known for providing or endorsing tasty food options specifically? I am not so sure. Although the intention was to manipulate source credibility, several other product-related perceptions may have been inadvertently manipulated too.

Methods

For me, the methods suggest limitations to how the labels were selected and characterised before applying them in the research. There are also some areas where more detail is required.

1. How were the locations (A&E vs Reception) selected? Were these intentional to serve a larger purpose?

2. Was a power calculation of any sort conducted or estimated? Please do add this detail.

3. Criteria were used determine the products health status. But do the authors have any evidence that consumers would perceive the more/less healthy products in this way? The congruency between a product’s perceived and labelled healthiness have the potential to impact how a label is subsequently viewed across a variety of dimensions, such as expectations of taste quality, and liking.

4. All the 12 healthier and 11 less healthy products were sold for 1 pound. Was the relative difference between this value and the products’ original cost accounted for? A product perceived to be greater ‘value’ for money may be purchased more, regardless of the label. Please clarify.

5. There is an assumption that the NHS label is seen as a more credible health label for food than the basic ‘healthier choice’ label. What preliminary testing/piloting was conducted to confirm this, before the study began? Please describe this. If there was no pilot testing of these labels then this is an important limitation of the study. How can you be sure you manipulated credibility as intended? How do you know that the labels did not impact consumer beliefs in other ways?

Analysis & Results

The results are fairly succinct, but some clarifications on the analyses are necessary to determine how robust they were:

1. Was the distribution of the data considered? What distributions assumptions were applied to the analysis? The error bars in Fig 3 and 4 suggest inconsistent variation across the variables.

2. Was any correction (e.g. Bonferroni etc.) applied to account for multiple testing when the additional tests were conducted to interpret the various interaction effects? Please provide these details. A correction should be applied.

3. It would be helpful to signify significant/non-significant differences in Fig 3-4 with bar labels or asterix. It is also important to state what the error bars represent in the figure legends (SEM? 95% CIs?).

Discussion

Depending on responses to the above, the limitation section may need expanding. It would also be great to see more critique that incorporates the findings from consumer studies of healthier labels and their impact on product perceptions and choices.

Reviewer #2: It is an interesting study with unexpected results as stated by the authors.

On Page 3, in the Background section, why did you include results at the end of the section

On Page 9, 3rd paragraph, try to rephrase the sentence because it is complicated and hard to understand

The values in the figures 3 and 4 "Average sales" stand for number of people who bought the products or what do they stand for?

Can the brand name, taste of food, flavour, and previous experience with product play a role in their selection

6. PLOS authors have the option to publish the peer review history of their article (what does this mean?). If published, this will include your full peer review and any attached files.

Reviewer #1: No

Reviewer #2: **Yes: **Ahlam Badreldin El Shikieri

---

## [Author Response · Author response to Decision Letter 0]

8 Nov 2023

Reviewer 1

We are very grateful for the helpful and considered comments provided by the Reviewer.

Reviewer #1: Introduction

Introduction is clearly written and succinct. A couple of points to consider:

1. The authors contextualise the main content of the introduction by sharing they found evidence for a label effect opposite to the one intended (so-called ‘backfire’ effect, where label credibility appears to increase unhealthy food sales in a particular context, rather than health food sales as intended). The possibility of backfire effects are outlined in the introduction, but it is not clear whether these effects were truly anticipated (or not) when the study was being set up. If they were anticipated, I would make sense to describe this in the introduction, but I would have expected more effort put into characterising how possible consumers are likely to perceive the various labels, prior to them being placed in the vending machines. See also my comment on the methods below. However, if the ‘backfire’ effects where not anticipated when the study was designed, it would be better to focus on this as a critique in the discussion.

- The discussion of backfire effects has now been moved to the discussion, with the results withheld until the Results / G.D. This reflects how the backfire effects were indeed unanticipated. The introduction now also features an expanded discussion of the justifications for exploring source credibility as a means of enhancing health labels, with the lack of exploration of the ways that these labels could shape other attitudes addressed in the limitations section of the G.D. 

2. Related to the above, although there may not be a lot of literature investigating source credibility effects in the food choice domain, there are quite a number of studies attempting to characterise the impact of ‘healthier’/ ‘lighter’/ ‘lower calorie’ labels and ‘healthier’ perceptions on consumer beliefs, including willingness to pay, and food choice, portion decisions and even intake. The body of evidence broadly indicates that ‘healthier’ labels/perceptions can impact consumer behaviours in a number of ways, and not always as intended. For example, ‘healthier’ perceptions are sometimes associated with poorer taste quality, which can reduce consumers interest in selecting these products. It would be good to see some of this consumer-focused literature considered in the paper. The NHS is sited as a trusted source of health information, but is it known for providing or endorsing tasty food options specifically? I am not so sure. Although the intention was to manipulate source credibility, several other product-related perceptions may have been inadvertently manipulated too.

-We now discuss these limitations in the G.D. 

Methods

For me, the methods suggest limitations to how the labels were selected and characterised before applying them in the research. There are also some areas where more detail is required.

1. How were the locations (A&E vs Reception) selected? Were these intentional to serve a larger purpose?

-The locations were selected for pragmatic rather than scientific purposes. We have now stated this in the method section. We highlight in the GD that a limitation of the study is the number and diversity of vending locations. 

2. Was a power calculation of any sort conducted or estimated? Please do add this detail.

- For practical reasons we were only able to conduct the study over a 90-day period. We have now stated this in the method section. Given this constraint, a priori power calculations were immaterial because we collected as much data as possible within the period allowed. In the GD, we discuss how a longer time period and a greater number of vending machines and locations would have elevated the power of the study. Furthermore, there is no accepted method of computing power when the data is being analysed using mixed effects models with two random effects (time and product type, as we used). 

3. Criteria were used determine the products health status. But do the authors have any evidence that consumers would perceive the more/less healthy products in this way? The congruency between a product’s perceived and labelled healthiness have the potential to impact how a label is subsequently viewed across a variety of dimensions, such as expectations of taste quality, and liking.

- A very interesting and valid point – we now address this in the G.D. 

4. All the 12 healthier and 11 less healthy products were sold for 1 pound. Was the relative difference between this value and the products’ original cost accounted for? A product perceived to be greater ‘value’ for money may be purchased more, regardless of the label. Please clarify.

- Random intercepts for products were included in the LME model. Thus if any one product was bought more often than the others due to say, extra perceived value, the extra variability across products that this introduced would have been taken into account by the model. 

5. There is an assumption that the NHS label is seen as a more credible health label for food than the basic ‘healthier choice’ label. What preliminary testing/piloting was conducted to confirm this, before the study began? Please describe this. If there was no pilot testing of these labels then this is an important limitation of the study. How can you be sure you manipulated credibility as intended? How do you know that the labels did not impact consumer beliefs in other ways?

- We have included a section on this in the GD.

Analysis & Results

The results are fairly succinct, but some clarifications on the analyses are necessary to determine how robust they were:

1. Was the distribution of the data considered? What distributions assumptions were applied to the analysis? The error bars in Fig 3 and 4 suggest inconsistent variation across the variables.

- The distribution of the data was considered. As stated in Footnote 1, one possibility would have been to run a logistic regression on the assumption that the data involved each person making a binary choice (i.e. a binary distribution). However, we cannot run a logistic regression because it is not the case that each person was randomly allocated to a treatment condition. Instead, we have people grouped together in a single time window, and we have randomly allocated time window to condition and averaged over individual choices. A logistic regression where each purchase is considered independent would inflate the degrees of freedom.

- There reviewer notes that the figure implies there is heterogeneity of variances across conditions. However, there are no significant differences in variability across conditions according to Levene’s test of equality of variances: all p’s > 0.05 when comparing the variability of one condition against the variability of another.

- One of the advantages of linear mixed effects models (as we used) are that they are robust to violations of assumptions such as heterogeneity of variance across conditions, e.g. Schielzeth et al. (2020; “Robustness of linear mixed‐effects models to violations of distributional assumptions”. Methods in Ecology and Evolution, cited 590 times) write, “We conclude that mixed-effects models are largely robust even to quite severe violations of model assumptions.”

2. Was any correction (e.g. Bonferroni etc.) applied to account for multiple testing when the additional tests were conducted to interpret the various interaction effects? Please provide these details. A correction should be applied.

- There is no correction applied but the crucial interactions in the omnibus analysis (p = .0031; p = .019 respectively) means that there have to be significant differences when simple effects are considered i.e. in the additional tests referred to by the reviewer. If we were to make corrections and those corrections removed the significant findings at the simple effects level, we would not be able to interpret the significant interactions in the omnibus analysis.

3. It would be helpful to signify significant/non-significant differences in Fig 3-4 with bar labels or asterix. It is also important to state what the error bars represent in the figure legends (SEM? 95% CIs?).

- Error bars are now stated in the figure legend (SEM). We don’t think asterisks on the figure signifying significance would be helpful in this case because there are many significant effects and the figure would become overloaded with asterisks, and many of the interesting effects are main effects or interactions that are not easily illustrated on a figure. 

Discussion

Depending on responses to the above, the limitation section may need expanding. It would also be great to see more critique that incorporates the findings from consumer studies of healthier labels and their impact on product perceptions and choices.

- We have now expanded the limitations section and moved discussion of the backfire effect from the introduction to the GD.

Reviewer 2

Reviewer #2: It is an interesting study with unexpected results as stated by the authors.

- We thank the reviewer for their kind words.

On Page 3, in the Background section, why did you include results at the end of the section

- We have removed the statement of results.

On Page 9, 3rd paragraph, try to rephrase the sentence because it is complicated and hard to understand

- We’re not sure which sentence the reviewer is referring to in the paragraph but we have rephrased where we feel it is appropriate.

The values in the figures 3 and 4 "Average sales" stand for number of people who bought the products or what do they stand for?

- We have made this clearer in the figure legends. The figures are average number of product sales per six-day time period, e.g. in the Lighter Choices, Healthy condition, there were 88 sales per six-day period, on average.

Can the brand name, taste of food, flavour, and previous experience with product play a role in their selection

It is likely that they do. However, the products were consistent across labelling conditions so any effects of individual products would not bias findings. 

- We also included product random effects and so variability across products is taken into account in the analysis.

---

## [Decision Letter · Decision Letter 1]

20 Dec 2023

Evaluating source credibility effects in health labelling using vending machines in a hospital setting

PONE-D-23-00303R1

Dear Dr. Bott,

We’re pleased to inform you that your manuscript has been judged scientifically suitable for publication and will be formally accepted for publication once it meets all outstanding technical requirements.

Kind regards,

Ali B. Mahmoud, Ph.D.

Academic Editor

PLOS ONE

Additional Editor Comments (optional):

Reviewers' comments:

Reviewer's Responses to Questions

**Comments to the Author**

1. If the authors have adequately addressed your comments raised in a previous round of review and you feel that this manuscript is now acceptable for publication, you may indicate that here to bypass the “Comments to the Author” section, enter your conflict of interest statement in the “Confidential to Editor” section, and submit your "Accept" recommendation.

Reviewer #2: All comments have been addressed

2. Is the manuscript technically sound, and do the data support the conclusions?

Reviewer #2: Yes

3. Has the statistical analysis been performed appropriately and rigorously? 

Reviewer #2: Yes

4. Have the authors made all data underlying the findings in their manuscript fully available?

Reviewer #2: Yes

5. Is the manuscript presented in an intelligible fashion and written in standard English?

Reviewer #2: Yes

6. Review Comments to the Author

Reviewer #2: None. All comments have been raised. The authors revised point-by-point and corrected what is requested, accordingly. There are no further comments

7. PLOS authors have the option to publish the peer review history of their article (what does this mean?). If published, this will include your full peer review and any attached files.

Reviewer #2: **Yes: **Ahlam Badreldin El Shikieri

---

## [Editor Report · Acceptance letter]

10 Feb 2024

PONE-D-23-00303R1 

PLOS ONE

Dear Dr. Bott, 

I'm pleased to inform you that your manuscript has been deemed suitable for publication in PLOS ONE. Congratulations! Your manuscript is now being handed over to our production team.

Kind regards, 

on behalf of

Dr. Ali B. Mahmoud 

Academic Editor

PLOS ONE